# Snake C-Type Lectins Potentially Contribute to the Prey Immobilization in *Protobothrops mucrosquamatus* and *Trimeresurus stejnegeri* Venoms

**DOI:** 10.3390/toxins12020105

**Published:** 2020-02-06

**Authors:** Huiwen Tian, Ming Liu, Jiameng Li, Runjia Xu, Chengbo Long, Hao Li, James Mwangi, Qiumin Lu, Ren Lai, Chuanbin Shen

**Affiliations:** 1Life Sciences College of Nanjing Agricultural University, Nanjing 210095, Jiangsu, China; 2017116051@njau.edu.cn (H.T.); 2017116055@njau.edu.cn (J.L.); 2Department of Molecular and Cell Biology, School of Life Sciences, University of Science and Technology of China, Anhui 230027, Hefei, China; qhdming@mail.ustc.edu.cn; 3Key Laboratory of Bioactive Peptides of Yunnan Province/Key Laboratory of Animal Models and Human Disease Mechanisms of Chinese Academy of Sciences, Kunming Institute of Zoology, Kunming 650223, Yunnan, China; 2016116040@njau.edu.cn (R.X.); longchengbo@mail.kiz.ac.cn (C.L.); lihao@mail.kiz.ac.cn (H.L.); jams@mail.kiz.ac.cn (J.M.); 4University of Chinese Academy of Sciences, Beijing 100009, China; 5Sino African Joint Research Center, CAS, Kunming Institute of Zoology, Kunming 650223, Yunnan, China

**Keywords:** snake venom, C-type lectin-like proteins, snaclecs, platelet and thrombocyte, cerebral ischemia, locomotor activity

## Abstract

Snake venoms contain components selected to immobilize prey. The venoms from Elapidae mainly contain neurotoxins, which are critical for rapid prey paralysis, while the venoms from Viperidae and Colubridae may contain fewer neurotoxins but are likely to induce circulatory disorders. Here, we show that the venoms from *Protobothrops mucrosquamatus* and *Trimeresurus stejnegeri* are comparable to those of *Naja atra* in prey immobilization. Further studies indicate that snake C-type lectin-like proteins (snaclecs), which are one of the main nonenzymatic components in viper venoms, are responsible for rapid prey immobilization. Snaclecs (mucetin and stejnulxin) from the venoms of *P. mucrosquamatus* and *T. stejnegeri* induce the aggregation of both mammalian platelets and avian thrombocytes, leading to acute cerebral ischemia, and reduced animal locomotor activity and exploration in the open field test. Viper venoms in the absence of snaclecs fail to aggregate platelets and thrombocytes, and thus show an attenuated ability to cause cerebral ischemia and immobilization of their prey. This work provides novel insights into the prey immobilization mechanism of Viperidae snakes and the understanding of viper envenomation-induced cerebral infarction.

## 1. Introduction

Snake venoms are complex cocktails of bioactive peptides and proteins that immobilize or digest prey [1]. There are more than 420 species of venomous snakes living on the earth [2], and variations in venom composition are common among these snakes [1,3]. Venomous snakes are classified into four families: Viperidae, Elapidae, Atractaspididae and Colubridae [4]. The venoms from elapids usually contain a high level of neurotoxins such as three-finger toxins (3FTxs) and phospholipases A2 (PLA2), that lead to the rapid paralysis of their prey [5,6,7]. In contrast, neurotoxins are less abundant in the venoms from vipers and colubrids, and their envenomation is often associated with hemorrhage and circulatory disorders [3,4]. It is unknown if the viper venoms have a comparable prey immobilization efficiency to that of Elapidae venoms and what their underlying mechanisms are.

Snake C-type lectin-like proteins (snaclecs) are mainly expressed in the venoms of vipers and colubrids [4,5]. The available data indicates that snaclecs may be one of the most abundant nonenzymatic group of proteins in the venoms [8,9,10,11,12,13]. Snalecs usually have a heterodimeric structure with α and β subunits, which are often oligomerized to form protein multimers, and have evolved to bind a wide range of physiologically important proteins such as GPIb, GPVI and integrins on mammal platelets [13,14,15,16]. In this study, we investigate the effects of viper snaclecs on prey immobilization and the mechanisms underlying them.

## 2. Results

### 2.1. Snaclecs Can Rapidly Immobilize Prey

The intraperitoneal injection of crude venom from *P. mucrosquamatus* or *T. stejnegeri* induced a quick loss of the righting reflex similar to the crude venom from *N. atra*, as shown in Table 1. Comparatively speaking, venoms from vipers and colubrids may contain fewer neurotoxins but are rich in snaclecs [4,5,14]. We therefore speculate that snaclecs such as mucetin or stejnulxin may be capable of immobilizing prey. The intraperitoneal injection of purified mucetin or stejnulxin reduced animal exploratory behavior and locomotor activity in the open field test in a concentration-dependent manner, as shown in Figure 1. The average travel distances of pheasant chicks (*Phasianus colchicus*) were reduced from 13.39 m to 1.95 and 1.32 m over a 10-min period with an increasing dose of mucetin or stejnulxin, respectively, as shown in Figure 1a. Mice (*Mus musculus*) were more active in the open field test, but the snaclecs showed a similar trend in reducing their spontaneous locomotor behavior. The average travel distances of mice were reduced from 3.40 m to 0.61 and 0.55 m over a 3-min period with an increasing dose of mucetin or stejnulxin, respectively, as shown in Figure 1b. Mice and birds are common snake prey belonging to different classes; however, mucetin and stejnulxin significantly reduced the locomotor activity of these animals indistinguishably, suggesting that snaclecs may be broad-spectrum toxins that help snakes to immobilize prey.

### 2.2. Snaclecs are Critical for Viper Venom Induced Prey Paralysis

To further investigate the role of mucetin and stejnulxin in viper envenomation-induced prey immobilization, we compared the activity of crude viper venoms both in the presence and absence of the snaclecs. As shown in Figure 2a, the average travel distances of pheasant chicks were significantly reduced from 13.39 m to 2.25 and 1.51 m by the crude venoms of *P. mucrosquamatus* and *T. stejnegeri*, respectively. Pheasant chicks treated with the venoms lacking mucetin or stejnulxin showed obviously extended travel distances, as shown in Figure 2a. A similar phenomenon was observed in mice, as shown in Figure 2b.

### 2.3. Snaclecs Induce Cerebral Ischemia

The intraperitoneal injection of the viper crude venoms or the snaclecs (mucetin or stejnulxin) at 400 μg/kg significantly reduced the cerebral blood flow of pheasant chicks, as shown in Figure 3a, as well as mice, as shown in Figure 3b, as monitored by a laser-speckle blood flow imaging system over a 10-min period. However, these crude venoms showed a much-attenuated effect on cerebral blood flow with the removal of mucetin or stejnulxin. Moreover, injection of the crude venoms from *N. atra*, which are considered to lack snaclecs, did not affect cerebral blood flow.

### 2.4. Snaclecs Activate Thrombocytes or Platelets 

Thrombocytes vary considerably among birds and mammals in terms of their cellular structure and functions, but they all share the identical function of clumping together quickly to form clots and thus preventing blood loss after trauma [17]. Abnormal activation of thrombocytes, such as platelet aggregation, will obstruct cerebral microcirculation and lead to cerebral infarction [18,19]. The available data indicate that mammalian platelets are the target of snaclecs [16,20,21], though whether these snaclecs have an effect on avian thrombocytes has never been investigated before. We compared the thrombocyte aggregation activity of the snaclecs as well as the crude viper venom both in the presence and absence of mucetin or stejnulxin, as shown in Figure 4. The crude venoms (CV) from *P. mucrosquamatus* and *T. stejnegeri* rather than *N. atra* potently induced thrombocyte aggregation at 2 μg/mL. The viper venoms in the absence of mucetin (CV-mu) or stejnulxin (CV-st) did not aggregate the thrombocytes or platelets at the same concentration. Further assays indicated that the purified snaclecs (mucetin or stejnulxin) showed a stronger ability to initiate aggregation than the crude venoms, as shown in Figure 4.

## 3. Discussion

Elapid envenomation usually leads to neurotoxicity because of the high content of three-finger toxins (3FTxs) and phospholipase A2 (PLA2) [4], while viper venoms rarely contain 3FTxs [5]. PLA2 also exists in *P. mucrosquamatus* and *T. stejnegeri* venoms [22,23], and in the latter shows an inhibitory effect on platelet aggregation [23]. Despite the presence of PLA2 in viper venoms that may lead to muscle paralysis, sensitivity to this toxin seems to be species-specific [24]. The injection of snake PLA2 does not immobilize prey immediately because there is always a minimum interval of about one hour between the injection and muscle paralysis, probably due to the mode of action but irrespective of the concentration [25,26]. In this study, we show that the venoms from *P. mucrosquamatus* and *T. stejnegeri* are comparable to those of Elapidae in prey immobilization, as shown in Table 1, and that the snaclecs may help vipers to rapidly immobilize and subdue prey by inducing platelet aggregation to impair blood circulation.

Viper snaclecs are one of the most abundant nonenzymatic group of proteins in some Viperidae venoms [8,9,10,11,12]. The Snaclecs reduced the exploratory behavior and locomotor activity of pheasant chicks as well as mice within 5 min of injection in a concentration-dependent manner, as shown in Figure 1. This suggests that the snaclecs may help vipers to immobilize their prey quickly and efficiently. According to the purification profile, we found that the content of platelet-activating snaclecs in these viper venoms are 5–10% (data not shown). It is worth noting that the purified mucetin and stejnulxin showed a stronger ability to aggregate thrombocytes/platelets than the crude venoms in vitro, as shown in Figure 4, but they are comparable with the crude venoms in animal experiments at the same concentration, as shown in Figure 1, Figure 2 and Figure 3. This is probably because crude viper venoms may contain components such as metalloproteinases and hyaluronidase that favor the spread of the snaclecs in the tissue and in the circulatory system [27]. Alternately, there may also be unknown factors in the crude venoms that are helpful for prey immobilization via inducing cardiovascular collapse and prolonged hypotension [28].

Despite the presence of anti-platelet proteins or peptides such as disintegrins in snake venoms [29,30], the crude viper venoms studied here showed a strong ability to induce thrombocytes/platelets aggregation, as shown in Figure 4. Given that the thrombocytes/platelets were not aggregated by the venoms that lacked snaclecs, the snaclecs (i.e., mucetin and stejnulxin) are the likely to be the major components that activate the aggregation of thrombocytes/platelets in the viper venoms. As a cocktail that contains high levels of proteins, snake venoms are metabolically expensive to produce [31]; however, the ability of viper venoms to impair the circulatory system of their prey (as well as humans) [4,13,14] is very efficient. Relatively fewer molecules are necessary to efficiently activate rather than inhibit thrombocyte/platelet aggregation due to the number of the receptors on the membrane and the triggered downstream cascade reactions [14]; thus, thrombocyte-activating snaclecs may provide key evidence to support the venom-optimization hypothesis [31].

The effects of snaclecs on the circulatory system were further confirmed by inducing acute cerebral ischemia. As illustrated in Figure 3, the intraperitoneal injection of the crude venoms from *P. mucrosquamatus* or *T. stejnegeri* as well as mucetin or stejnulxin significantly reduced the cerebral blood flow in both the pheasant chicks and the adult mice within 5 min, while the cerebral blood flow in mice treated by the *N. atra* venom or the viper venoms lacking snaclecs was comparable with the normal saline group. This suggested that mucetin and stejnulxin may be the main components for these venom-induced cerebral infarctions. Birds and small rodent mammals are common prey for snakes [32,33]. Cerebral infarction caused by snaclecs likely facilitates vipers to immobilize and capture their prey by inducing motor disability in animals, as illustrated in Figure 1 and Figure 2. Despite a low occurrence, available data indicate that viper envenomation, including *T. stejnegeri* [34], leads to acute cerebral infarction in humans [4,35,36,37,38,39]; however, experiments performed so far have not clarified the components responsible for viper envenomation-induced cerebral infarction in the venom. We suggest that Snaclecs may be the key component for viper envenomation-induced cerebral infarction. Our current finding that viper snaclecs induce cerebral infarction provides new insight into our understanding of viper envenomation-induced cerebral infarction.

## 4. Material and Methods

### 4.1. Venom Collection and Toxins Purification

The crude venoms from *P. mucrosquamatus* or *T. stejnegeri* were collected in Jiangxi province of China. The purification and identification of mucetin and stejnulxin were carried out as previously described [15,16], and their amino acid sequences of α and β subunits are shown in Appendix A. The purity of the snaclecs was determined by SDS-PAGE and is shown in Appendix A. FPLC (Fast Protein Liquid Chromatography) was used to remove mucetin or stejnulxin from the crude venoms, while the crude venoms in the absence of mucetin (CV-mu) and stejnulxin (CV-st) were made by combining different components, without mucetin and stejnulxin.

### 4.2. The Locomotor Activity in an Open Field Test

The open field test is a standard test apparatus used to measure animal locomotion activity and exploration behavior [40]. An open field test was performed according to the previous method described [41]; briefly, pheasant chicks (190–210 g) and BALB/c mice (20–22 g) of either sex were used in this experiment. Experimental protocol (SMKX2017027) using animals in this work was subjected to prior review by the Animal Care and Use Committee at Kunming Institute of Zoology, Chinese Academy of Sciences in December 2017. The dimensions of the open-field arena for chicks were 1.5 m in diameter and 60 cm in height, while the chamber for the mice was 50 × 50 cm with the same height. The animals were gently placed individually in the open-field arena, and the behaviors of the animals were monitored by an automated infrared tracking system (Noldus version 8.0, ZS Dichuang, Beijing, China, 2014) one minute after the injection of the toxins. The total distance travelled within 10 min and 3 min were recorded, respectively, for chicks and mice, and further analyzed with GraphPad prism (Version 8.0, GraphPad Software, San Diego, CA, USA, 2018).

### 4.3. Continuous Measurement of Cerebral Cortex Blood Flow

Pheasant chicks (190–210 g) and BALB/c mice (20–22 g) of either sex were anesthetized by isoflurane inhalation with an anesthesia respirator (R540IP, RWD Life Science). The head was fixed, and the scalp was cut longitudinally to expose the skull of the mouse. A gentle saline drip over the exposed surgical opening prevented dehydration of the skull. The blood flow in the cerebral cortex of the animals was monitored by a laser-speckle blood flow imaging system (Version 2.0, RFLSI Pro, RWD Life Science, Shenzhen, China, 2017) before and after the injection of the toxins.

### 4.4. Platelet/Thrombocyte Isolation and Stimulation 

Thrombocytes were isolated as previously described, but with some modifications [42]. Briefly, blood was collected from the wing vein of adult pheasant chick and put into a 50 mL sterile polystyrene tube containing 1mL 10% EDTA solution. The blood sample was diluted 1:1 with Hank’s equilibrium salt solution (HBSS) without Ca^2+^ and Mg^2+^. Diluted blood samples (6 mL) were added to the upper layer of the lymphocyte separation medium (density = 1.077 g/mL, GE Healthcare) and centrifuged at 1700× *g* for 40 min at room temperature. The thrombocytes in the intermediate layer were collected and washed with Tyrode’s buffer A (137 mM NaCl, 2 mM KCl, 0.3 mM NaH_2_PO_4_, 12 mM NaHCO_3_, 5.5 mM glucose, 0.35% BSA, 1 mM MgCl_2_ and 0.2 mM EDTA, pH 6.5) by centrifugation at 450× *g* for 5 min at room temperature. The platelets were isolated from the mice with differential centrifugation. Briefly, platelet-rich plasma (PRP) was isolated from the citrated whole blood by centrifuging at 100× *g* for 5 min at room temperature, then the platelets were pelleted at 500× *g* for 5 min and washed with Tyrode’s buffer A. The thrombocytes or platelets were suspended in Tyrode’s buffer B (137 mM NaCl, 2 mMKCl, 0.3 mM NaH_2_PO4, 12 mM NaHCO_3_, 5.5 mM glucose, 0.35% BSA and 2 mM CaCl_2_, pH 7.4) for further use. Aggregation was elicited by the addition of the toxins to platelets/thrombocytes and stirred at 1000 rpm for 5 min at 37 °C in a four-channel aggregometer (LBY-NJ4, Techlink, Beijing, China).

### 4.5. Statistical Analysis

A nonparametric test with a Dunn’s multiple comparison test was used to indicate the statistical significant differences between the groups. Analyses were performed with GraphPad Prism 8 software. The results were reported as mean ± SD with significance accepted at *p* < 0.05.

## Figures and Tables

**Figure 1 toxins-12-00105-f001:**
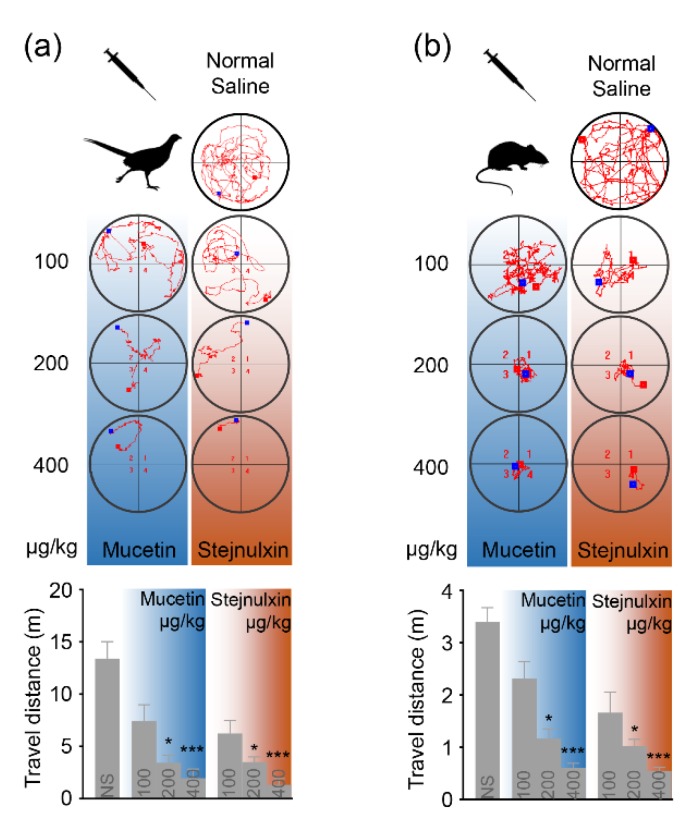
The effect of mucetin and stejnulxin on the locomotor activity and exploration of pheasant chicks and mice in the open field test. The animals were placed in the open-field arena 1 min after the injection of mucetin and stejnulxin. The distinct exploration paths of the pheasant chicks (**a**) and the mice (**b**) within 10 and 3 min, respectively, were recorded by an automated infrared tracking system. A nonparametric test with a Dunn’s multiple comparison test were used to indicate the statistical significant differences between the groups (N = 6, * *p* < 0.05, *** *p* < 0.001).

**Figure 2 toxins-12-00105-f002:**
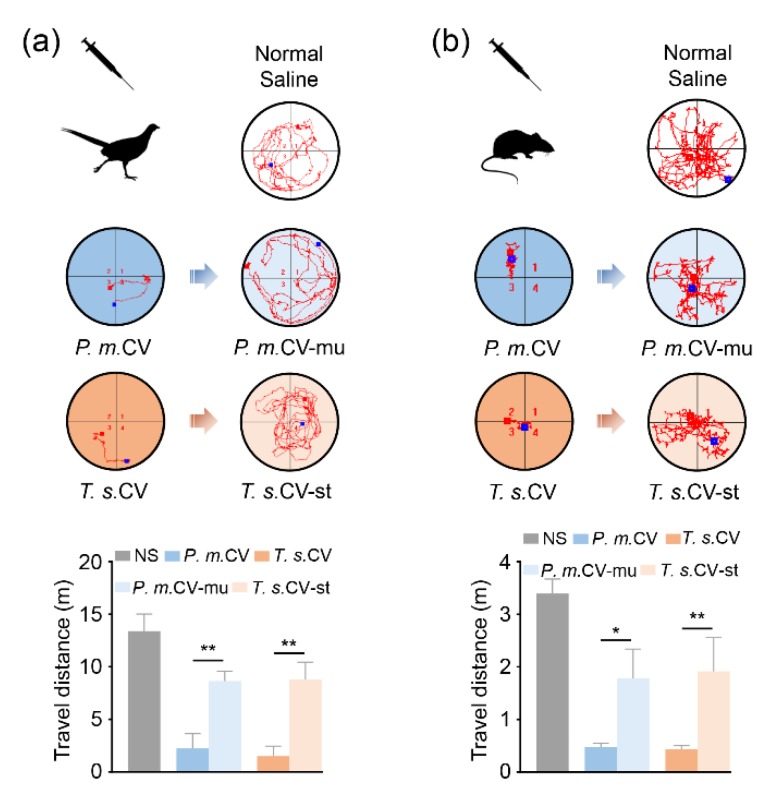
Mucetin and stejnulxin are critical for viper venom-induced prey immobilization. The animals were placed in the open-field arena one minute after the injection of the crude venoms (CV) from *P. mucrosquamatus* (*P.m.*) and *T. stejnegeri* (*T.s.*) or the crude venoms in the absence of mucetin (CV-mu) and stejnulxin (CV-st; 400 μg/kg). The distinct exploration paths of pheasant chicks (**a**) and mice (**b**) within 10 and 3 min, respectively, were recorded by an automated infrared tracking system. A nonparametric test with a Dunn’s multiple comparison test were used to indicate statistical significant differences between groups (N = 6, * *p* < 0.05, ** *p* < 0.01).

**Figure 3 toxins-12-00105-f003:**
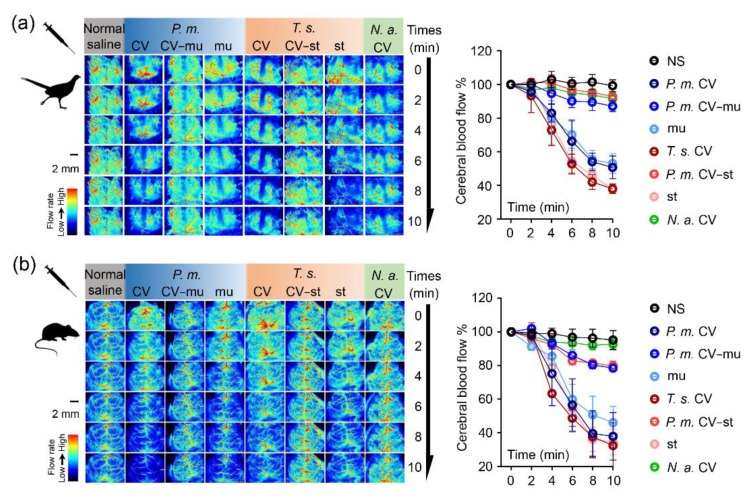
Mucetin- and stejnulxin-induced cerebral infarction in pheasant chicks and mice. Representative images and cerebral cortex blood flow quantification of pheasant chicks (**a**) and mice (**b**) treated with the purified mucetin (mu) and stejnulxin (st) or the crude venoms (CV) from *P. mucrosquamatus* (*P.m.*) and *T. stejnegeri* (*T.s.*), with or without mucetin (CV-mu) and stejnulxin (CV-st), at a concentration of 400 μg/kg. A nonparametric test with a Dunn’s multiple comparison test was used to indicate the statistical significant differences between groups (n = 3).

**Figure 4 toxins-12-00105-f004:**
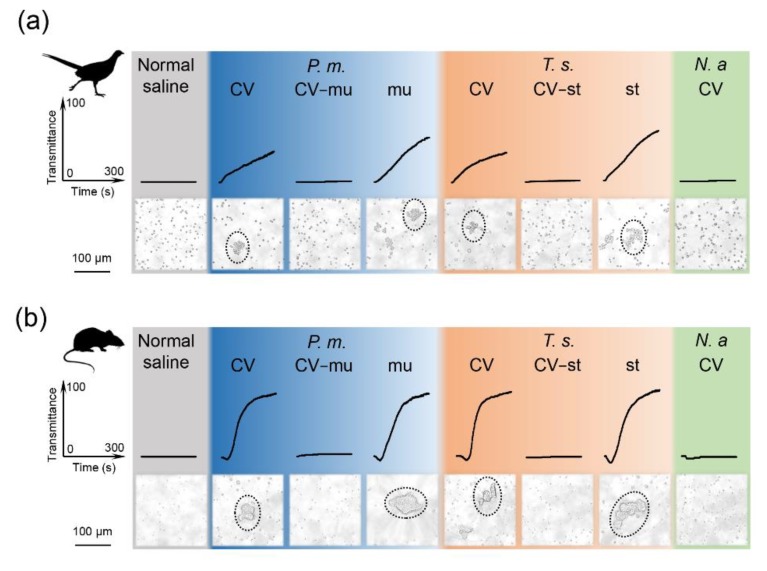
Mucetin- and stejnulxin-induced aggregation of thrombocytes or platelets from pheasant chicks and mice. The aggregation of pheasant chick thrombocytes (**a**) and mouse platelets (**b**) induced by the purified mucetin (mu) and stejnulxin (st) at a concentration of 1.0 μg/mL or the crude venoms (CV) from *P. mucrosquamatus* (*P.m.*) and *T. stejnegeri* (*T.s.*), with or without mucetin (CV-mu) and stejnulxin (CV-st), at a concentration of 2.0 μg/mL. The aggregates of the thrombocytes or platelets were visualized by light microscopy.

**Table 1 toxins-12-00105-t001:** The percentage of prey species to lose their righting reflex after intraperitoneal injection of crude snake venom for 2.5, 5 and 10 min.

Prey Species	Crude Venom	Percentage to Lose Righting Reflex (%)		
	(2 mg/kg)	2.5 min	5 min	10 min
Pheasant chicks	*P. mucrosquamatus*	5%	30%	75%
(*P. colchicus*)	*T. stejnegeri*	10%	40%	90%
N = 20 per group	*N.atra*	5%	40%	85%
Adult mice	*P. mucrosquamatus*	30%	70%	100%
(*M. musculus*)	*T. stejnegeri*	40%	85%	100%
N = 20 per group	*N. atra*	35%	80%	100%

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
