# Peer review of "Snake C-Type Lectins Potentially Contribute to the Prey Immobilization in Protobothrops mucrosquamatus and Trimeresurus stejnegeri Venoms"

_toxins, 2020, doi:10.3390/toxins12020105_

Round 1
Reviewer 1 Report
This paper deals with the novel and interesting observation that platelet aggregation-inducing snaclecs present in certain viperid venoms may serve to rapidly immobilise prey. The evidence is sound and deserves publication. However, before recommending this, revision is needed to address the following points:
Line 4: "Snake venoms contain components SELECTED to immobilize prey.." Language requires revision, i.e. "A complex cocktailS", "enemy-immobilizing efficiency"... Line 30: "Venoms from vipers and colubrids lack neurotoxins..."; Line 43: " 4. Although viper venoms lack neurotoxins..", Line 46: "... venoms from vipers and colubrids lack neurotoxins ", really? V. ammodytes, C. scutulatus, C. tigris, B. irregularis... Lines 35-37: "Available data indicate that they are one of the main components which account for 10-20% of viper venoms, and may be the most abundant non-enzymatic group of proteins in the venoms [6-11]" Please, revise! What was the rationale for selecting the doses of venom and isolated proteins? The purity of mucetin and stejnulxin should be illustrated (SDS-PAGE and or RP-HPLC plus determination of molecular mass by MS). Lines 68-69: "... we compared the activity of crude viper venoms both in the presence and absence of the snaclecs". How were the snaclecs removed from the venoms? Lines 82-83: " Intraperitoneal injection of viper crude venoms or snaclecs (mucetin or stejnulxin) at 400 μg/kg significantly reduced the cerebral blood flow of pheasant chicks..". Please, relate this dose of snaclecs to the amount of whole venom that contained it. Also, relate the amount of venom containing that quantity of snaclecs with the mean venom yield of the snake and the LD50 of its venom. Lines 122-123: "... snaclecs may help vipers to rapidly immobilize and subdue prey by inducing platelet aggregation to impair blood circulation". There are other platelet-inducing toxins in viperid venom, including PLA2 molecules. It would be relevant to check if these toxins mirror the prey-immobilizing activity of mucetin and stejnulxin... Lines 124-125: " Viper snaclecs, which are the most abundant non-enzymatic group of proteins in SOME Viperidae venoms..." Lines 133-134: "... it is wise for a snake to destroy its opponents’ thrombocytes or platelets.." Such teleologic expression are not appropriatein a scientific paper... Lines 137-138: "Thrombocyte-activating snaclecs may be one of the best evidence to support the 137 venom-optimization hypothesis [28]". The meaning of this statement is really enigmatic.. Lines 147-149: "Our current finding that viper snaclecs have the function to induce cerebral infarction provides new insight into the treatment for viper envenomation". Please, clarify.Author Response
Dear Editor,
Thank you and your reviewers for the extremely helpful comments provided for our manuscript (toxins-695282).
We have studied the comments carefully and made a lot of modifications, and we hope that our responses may answer the questions that raised by the reviewers. Below are our point-by-point responses to reviewers’ comments:
Comments from Reviewer 1:
Line 4: "Snake venoms contain components SELECTED to immobilize prey.." Language requires revision, i.e. "A complex cocktailS", "enemy-immobilizing efficiency"...
Response: Thank you for your advices. Revisions had been made according to your suggestions in the newly uploaded version of this manuscript.
Line 30: "Venoms from vipers and colubrids lack neurotoxins..."; Line 43: " 4. Although viper venoms lack neurotoxins..", Line 46: "... venoms from vipers and colubrids lack neurotoxins ", really? V. ammodytes, C. scutulatus, C. tigris, B. irregularis...
Response: Thank you for your advice. Venoms from vipers and colubrids don’t lack neurotoxins, but relatively speaking, neurotoxins are less abundant in the venoms from vipers and colubrids compared with venoms from elapids. So we modified the sentence "Venoms from vipers and colubrids lack neurotoxins..." to “While neurotoxins are less abundant in the venoms from vipers and colubrids”; We deleted the sentence “Although viper venoms lack neurotoxins.." and changed the sentence “... venoms from vipers and colubrids lack neurotoxins“ to “venoms from vipers and colubrids may contain less neurotoxins” in the newly uploaded version of this manuscript.
Lines 35-37: "Available data indicate that they are one of the main components which account for 10-20% of viper venoms, and may be the most abundant non-enzymatic group of proteins in the venoms [6-11]" Please, revise!
Response: Thank you for your advice. We revised the sentence to “Available data indicate that snaclecs may be one of the most abundant non-enzymatic group of proteins in the venoms” in the newly uploaded version of this manuscript.
What was the rationale for selecting the doses of venom and isolated proteins?
Response: We found that the LD50 of the lyophilized powder of these venoms are 3 to 4 mg/kg via intraperitoneal injection. According to the purification profile, we found that the content of platelet-activating snaclecs in these viper venoms are 5-10%, so we inject 400 μg/kg isolated proteins in animal experiments. However, crude viper venoms contain components that favoring the spread of the toxins in the tissue and to the circulation, such as metalloproteinases and hyaluronidase. Or some hitherto unknown factors in the crude venoms may work in concert with snaclecs to activate thrombocytes/platelets. So snaclecs within the crude venoms maybe more toxic than the purified snaclecs in vivo. We added this information in the newly uploaded version of this manuscript.
The purity of mucetin and stejnulxin should be illustrated (SDS-PAGE and or RP-HPLC plus determination of molecular mass by MS).
Response: Thank you for your advice. We added the SDS-PAGE in the newly uploaded version of this manuscript.
Lines 68-69: "... we compared the activity of crude viper venoms both in the presence and absence of the snaclecs". How were the snaclecs removed from the venoms?
Response: Snaclecs was removed from the venoms by FPLC (Fast Protein Liquid Chromatography), mucetin or stejnulxin was determined by platelet aggregation assay and SDS-PAGE, and the rest of the venom components were combined together without mucetin or stejnulxin to generate the mucetin or stejnulxin depleted venoms.
Lines 82-83: " Intraperitoneal injection of viper crude venoms or snaclecs (mucetin or stejnulxin) at 400 μg/kg significantly reduced the cerebral blood flow of pheasant chicks..". Please, relate this dose of snaclecs to the amount of whole venom that contained it. Also, relate the amount of venom containing that quantity of snaclecs with the mean venom yield of the snake and the LD50 of its venom.
Response: Thank you for your advice. According to the purification profile, we found that the content of platelet-activating snaclecs in these viper venoms are 5-10%. We added this information in the newly uploaded version of this manuscript.
Lines 122-123: "... snaclecs may help vipers to rapidly immobilize and subdue prey by inducing platelet aggregation to impair blood circulation". There are other platelet-inducing toxins in viperid venom, including PLA2 molecules. It would be relevant to check if these toxins mirror the prey-immobilizing activity of mucetin and stejnulxin...
Response: Thank you for your constructive suggestion. Previous study have revealed that PLA2 from Trimeresurus stejnegeri may inhibit platelet aggregation. Also, injection of snake PLA2 will not immobilize prey immediately, because there is always a minimum interval of about one hour between injection and muscle paralysis, probably due to the mode of action, but irrespective of the concentration.
Lines 124-125: " Viper snaclecs, which are the most abundant non-enzymatic group of proteins in SOME Viperidae venoms..." Lines 133-134: "... it is wise for a snake to destroy its opponents’ thrombocytes or platelets.." Such teleologic expression are not appropriatein a scientific paper...
Response: Thank you for your advice. Revisions had been made according to your suggestions in the newly uploaded version of this manuscript.
Lines 137-138: "Thrombocyte-activating snaclecs may be one of the best evidence to support the 137 venom-optimization hypothesis [28]". The meaning of this statement is really enigmatic..
Response: Thank you for your advice. Snake venoms contain high level of proteins which is metabolically expensive, the high metabolic cost of venom leads to the prediction that venomous animals may have evolved strategies for minimizing venom expenditure.
Lines 147-149: "Our current finding that viper snaclecs have the function to induce cerebral infarction provides new insight into the treatment for viper envenomation". Please, clarify.
Response: Thank you for your advice. Acute cerebral infarction or stroke after viper bites are reported in humans. However, experiments performed so far have not clarified the components for viper envenomation-induced cerebral infarction in the venom. Snaclecs maybe the key components for viper envenomation-induced cerebral infarction. Antibodies against Snaclecs may alleviate viper envenomation-induced cerebral infarction.

Reviewer 2 Report
I reviewed the research article titled “Vipers rapidly immobilize prey by venom C-type lectin like toxins” which is based on an interesting and novel idea. The argument generated here is that the viper and colubrid venoms are less neurotoxic, hence they should have a mechanism of rapid prey immobilisation. The authors demonstrate that the c-type lectins aggregate platelets, likely leading to cerebral ischemia resulting immobilisation.
First, authors must be very careful about the taxonomy and the nomenclature of the snakes. The Brown spotted pit-viper was earlier placed in the genus Trimeresurus, but it is currently placed in the genus Protobothrops. (please read: Guo, P et. al. Multilocus phylogeography of the brown-spotted pitviper Protobothrops mucrosquamatus (Reptilia: Serpentes: Viperidae) sheds a new light on the diversification pattern in Asia. Molecular Phylogenetics and Evolution 2019, 133, 82–91). Therefore, the name T. mucrosquamatus must be corrected throughout the manuscript as P. mucrosquamatus.
The authors have appropriately proved experimentally the following,
Mucetin and stejnulxin, the two c-type lectins and the venoms of P. mucrosquamatus and stejnegeri inhibit the locomotor activity of mice and birds. mucetin and stejnulxin and the crude venoms of P. mucrosquamatus and stejnegeri activate thrombocytes. Mucetin and stejnulxin affect cerebral blood flow, inducing cerebral ischemia.Therefore, I support the authors’ claim that the two snake c-type lectins may potentially contribute to the prey immobilisation in snake venoms. However, it is beyond the scope of this study to claim this as “Vipers rapidly immobilize prey by venom C-type lectin like toxins” as shown in the title, as it is a clear exaggeration as the authors have observed the effect in only two species of old-world pit viper venoms.
Therefore, I suggest the authors to make the title to be more focused on their findings.
Further, it appears that the authors consider that the vipers do not have neurotoxins in their venoms as implied from their statements (example: “Although viper venoms lack neurotoxins they can quickly immobilize prey” – line 43). There are many examples for pre-synaptic and post-synaptic toxins from viperid venoms. There are post-synaptic neurotoxins in colubrid venoms as well. I suggest the authors to correct the above statement.
When reading the article, it appears to the reader that this c-type lectin mediated cerebral hypoperfusion mechainsms are the main mechanism that the vipers are evolved with in incapacitating prey. However, authors have not mentioned that the cardiovascular collapse and prolonged hypotension, which are two mechanisms that are used by snakes to rapidly immobilize the prey. Elapids and viperids both use these mechanisms (please read: Kakumanu et al. 2019 https://www.nature.com/articles/s41598-019-56643-0) . Therefore, it is noteworthy that these snakes have more potent and rapidly acting weapons to immobilize prey. Certainly c-type lectins may also could contribute, but they are not the only or major toxins that serves the purpose. Therefore, the authors should bring these into the discussion to make the discussion section rich.
Methods: Line 173 – There seems an error in citing the reference number 17. The ref. 17 is a review article and it does not describe methods of platelet isolation.
Author Response
Dear Editor,
Thank you and your reviewers for the extremely helpful comments provided for our manuscript (toxins-695282).
We have studied the comments carefully and made a lot of modifications, and we hope that our responses may answer the questions that raised by the reviewers. Below are our point-by-point responses to reviewers’ comments:
Comments from Reviewer 2:
First, authors must be very careful about the taxonomy and the nomenclature of the snakes. The Brown spotted pit-viper was earlier placed in the genus Trimeresurus, but it is currently placed in the genus Protobothrops. (please read: Guo, P et. al. Multilocus phylogeography of the brown-spotted pitviper Protobothrops mucrosquamatus (Reptilia: Serpentes: Viperidae) sheds a new light on the diversification pattern in Asia. Molecular Phylogenetics and Evolution 2019, 133, 82–91). Therefore, the name T. mucrosquamatus must be corrected throughout the manuscript as P. mucrosquamatus.
Response: Thank you for your advice. Revisions had been made according to your suggestions in the newly uploaded version of this manuscript.
The authors have appropriately proved experimentally the following,
Mucetin and stejnulxin, the two c-type lectins and the venoms of P. mucrosquamatus and stejnegeri inhibit the locomotor activity of mice and birds. mucetin and stejnulxin and the crude venoms of P. mucrosquamatus and stejnegeri activate thrombocytes. Mucetin and stejnulxin affect cerebral blood flow, inducing cerebral ischemia.
Therefore, I support the authors’ claim that the two snake c-type lectins may potentially contribute to the prey immobilisation in snake venoms. However, it is beyond the scope of this study to claim this as “Vipers rapidly immobilize prey by venom C-type lectin like toxins” as shown in the title, as it is a clear exaggeration as the authors have observed the effect in only two species of old-world pit viper venoms.
Therefore, I suggest the authors to make the title to be more focused on their findings.
Response: Thank you for your advice. We changed the title to “Snake c-type lectins contribute to the prey immobilization in Protobothrops mucrosquamatus and Trimeresurus stejnegeri venoms”. Revisions had been made according to your suggestions in the newly uploaded version of this manuscript.
Further, it appears that the authors consider that the vipers do not have neurotoxins in their venoms as implied from their statements (example: “Although viper venoms lack neurotoxins they can quickly immobilize prey” – line 43). There are many examples for pre-synaptic and post-synaptic toxins from viperid venoms. There are post-synaptic neurotoxins in colubrid venoms as well. I suggest the authors to correct the above statement.
Response: Thank you for your advice. Revisions had been made according to your suggestions in the newly uploaded version of this manuscript.
When reading the article, it appears to the reader that this c-type lectin mediated cerebral hypoperfusion mechainsms are the main mechanism that the vipers are evolved with in incapacitating prey. However, authors have not mentioned that the cardiovascular collapse and prolonged hypotension, which are two mechanisms that are used by snakes to rapidly immobilize the prey. Elapids and viperids both use these mechanisms (please read: Kakumanu et al. 2019 https://www.nature.com/articles/s41598-019-56643-0) . Therefore, it is noteworthy that these snakes have more potent and rapidly acting weapons to immobilize prey. Certainly c-type lectins may also could contribute, but they are not the only or major toxins that serves the purpose. Therefore, the authors should bring these into the discussion to make the discussion section rich.
Response: Thank you for your advice. We have brought this new information in the discussion section according to your suggestion in the newly uploaded version of this manuscript.
Methods: Line 173 – There seems an error in citing the reference number 17. The ref. 17 is a review article and it does not describe methods of platelet isolation.
Response: Thank you for your advice. Revisions had been made in the newly uploaded version of this manuscript.

Round 2
Reviewer 1 Report
Authors have adequately addressed my concerns to the original version of the manuscript.
Author Response
Thank you for the extremely helpful comments provided for our manuscript (toxins-695282).